# Carbon recovery dynamics following disturbance by selective logging in Amazonian forests

Camille Piponiot[1,2,3,4*], Plinio Sist[4], Lucas Mazzei[5], Marielos Peña-Claros[6], Francis E Putz[7], Ervan Rutishauser[8], Alexander Shenkin[9], Nataly Ascarrunz[10], Celso P de Azevedo[11], Christopher Baraloto[12], Mabiane França[11], Marcelino Guedes[13], Eurídice N Honorio Coronado[14], Marcus VN d'Oliveira[15], Ademir R Ruschel[5], Kátia E da Silva[11], Eleneide Doff Sotta[13], Cintia R de Souza[11], Edson Vidal[16], Thales AP West[7], Bruno Hérault[2*]

[1]Université de Guyane, UMR EcoFoG (Agroparistech, CNRS, Inra, Université des Antilles, Cirad), Kourou, French Guiana; [2]Cirad, UMR EcoFoG (Agroparistech, CNRS, Inra, Université des Antilles, Université de Guyane), Kourou, French Guiana; [3]CNRS, UMR EcoFoG (Agroparistech, Inra, Université des Antilles, Université de Guyane, Cirad), Kourou, French Guiana; [4]Cirad, UR Forests and Societies, Montpellier, France; [5]Embrapa Amazônia Oriental, Belém, Brazil; [6]Forest Ecology and Forest Management Group, Wageningen University, Wageningen, Netherlands; [7]Department of Biology, University of Florida, Gainesville, United States; [8]CarbonForExpert, Hermance, Switzerland; [9]Environmental Change Institute, University of Oxford, Oxford, United Kingdom; [10]Instituto Boliviano de Investigación Forestal, Santa Cruz, Bolivia; [11]Embrapa Amazônia Ocidental, Belém, Brazil; [12]Department of Biological Sciences, International Center for Tropical Botany, Florida International University, Miami, United States; [13]Embrapa Amapa, Macapa, Brazil; [14]Instituto de Investigaciones de la Amazonia Peruana, Iquitos, Peru; [15]Embrapa Acre, Rio Branco, Brazil; [16]Departamento de Ciências Florestais, University of São Paulo, Piracicaba, Brazil

*For correspondence: camille.
piponiot@gmail.com (CP); bruno.
herault@cirad.fr (BH)

Competing interests: The authors declare that no competing interests exist.

**Abstract** When 2 Mha of Amazonian forests are disturbed by selective logging each year, more than 90 Tg of carbon (C) is emitted to the atmosphere. Emissions are then counterbalanced by forest regrowth. With an original modelling approach, calibrated on a network of 133 permanent forest plots (175 ha total) across Amazonia, we link regional differences in climate, soil and initial biomass with survivors' and recruits' C fluxes to provide Amazon-wide predictions of post-logging C recovery. We show that net aboveground C recovery over 10 years is higher in the Guiana Shield and in the west (21 $\pm$3 Mg C ha$^{-1}$) than in the south (12 $\pm$3 Mg C ha$^{-1}$) where environmental stress is high (low rainfall, high seasonality). We highlight the key role of survivors in the forest regrowth and elaborate a comprehensive map of post-disturbance C recovery potential in Amazonia.

## Introduction

With on-going climate change, attention is increasingly drawn to the impacts of human activities on carbon (C) cycles (*Griggs and Noguer, 2002*), and in particular to the 2.1 $\pm$ 1.1 Pg C yr$^{-1}$ of C loss caused by various forms and intensities of anthropogenic disturbances in tropical forests

**eLife digest** The Amazon rainforest in South America is the largest tropical forest in the world. Along with being home to a huge variety of plants and wildlife, rainforests also play an important role in storing an element called carbon, which is a core component of all life on Earth. Certain forms of carbon, such as the gas carbon dioxide, contribute to climate change so researchers want to understand what factors affect how much carbon is stored in rainforests. Trees and other plants absorb carbon dioxide from the atmosphere and then incorporate the carbon into carbohydrates and other biological molecules. The Amazon rainforest alone holds around 30% of the total carbon stored in land-based ecosystems.

Humans selectively harvest certain species of tree that produce wood with commercial value from the Amazon rainforest. This "selective logging" results in the loss of stored carbon from the rainforest, but the loss can be compensated for in the medium to long term if the forest is left to regrow. New trees and trees that survived the logging grow to fill the gaps left by the felled trees. However, it is not clear how differences in the forest (for example, forest maturity), environmental factors (such as climate or soil) and the degree of the disturbance caused by the logging affect the ability of the forest ecosystem to recover the lost carbon.

Piponiot et al. used computer modeling to analyze data from over a hundred different forest plots across the Amazon rainforest. The models show that the forest's ability to recover carbon after selective logging greatly differs between regions. For example, the overall amount of carbon recovered in the first ten years is predicted to be higher in a region in the north known as the Guiana Shield than in the south of the Amazonian basin where the climate is less favorable.

The findings of Piponiot et al. highlight the key role the trees that survive selective logging play in carbon recovery. The next step would be to couple this model to historical maps of logging to estimate how the areas of the rainforest that are managed by selective logging shape the overall carbon balance of the Amazon rainforest.

(*Grace et al., 2014*). Among those disturbances, selective logging, i.e. the selective harvest of a few merchantable tree species, is particularly widespread: in the Brazilian Amazon alone, about 2 Mha yr$^{-1}$ were logged in 1999–2002 (*Asner et al., 2005*). The extent of selective logging in the Brasilian Amazon was equivalent to annual deforestation in the same period, and resulted in C emissions of 90 Tg C yr$^{-1}$ (*Huang and Asner, 2010*) which increased anthropogenic C emissions by almost 25% over deforestation alone (*Asner et al., 2005*). In contrast to deforested areas that are used for agriculture and grazing, most selectively logged forests remain as forested areas (*Asner et al., 2006*) and may recover C stocks (*West et al., 2014*). Previously logged Amazonian forests may thus accumulate large amounts of C (*Pan et al., 2011*), but this C uptake is difficult to accurately estimate, because while detecting selective logging from space is increasingly feasible (*Frolking et al., 2009*) (even if very few of the IPCC models effectively account for logging), directly quantifying forest recovery remains challenging (*Asner et al., 2009*; *Houghton et al., 2012*; *Goetz et al., 2015*). Studies based on field measurements (e.g. *Sist and Ferreira, 2007*; *Blanc et al., 2009*; *West et al., 2014*; *Vidal et al., 2016*), sometimes coupled with modeling approaches (e.g. *Gourlet-Fleury et al., 2005*; *Valle et al., 2007*) or airborne light detection and ranging (LiDAR) measurements (e.g. *Andersen et al., 2014*) have assessed post-logging dynamics at particular sites. Nonetheless, to our knowledge no spatially-explicit investigation of post-logging C dynamics at the Amazon biome scale is available.

C losses from selective logging are determined by harvest intensity (i.e. number of trees felled or volume of wood extracted) plus the care with which harvest operations are conducted, which affects the amount of collateral damage. After logging, C losses continue for several years due to elevated mortality rates of trees injured during harvesting operations (*Shenkin et al., 2015*). Logged forests may recover their aboveground carbon stocks (ACS) via enhanced growth of survivors and recruited trees (*Blanc et al., 2009*). Full recovery of pre-disturbance ACS in logged stands reportedly requires up to 125 years, depending primarily on disturbance intensity (*Rutishauser et al., 2015*). The underlying recovery processes (i.e. tree mortality, growth and recruitment) are likely to vary with the clear

geographical patterns in forest structure and dynamics across the Amazon Basin and Guiana Shield. In particular, northeast-southwest gradients have been reported for ACS (*Malhi and Wright, 2004*), net primary productivity (*Aragão et al., 2009*), wood density (*Baker et al., 2004*), and floristic composition (*ter Steege et al., 2006*). Such gradients coincide with climate and edaphic conditions that range from nearly a seasonal nutrient-limited in the northeast to seasonally dry and nutrient-rich in the southwest (*Quesada et al., 2012*). These regional differences in biotic and abiotic conditions largely constrain demographic processes that ultimately shape forest C balances.

Here we partition the contributions to post-disturbance ACS gain (from growth and recruitment of trees ≥20 cm DBH) and ACS loss (from mortality) of survivors and recruited trees to detect the main drivers and patterns of ACS recovery in forests disturbed by selective logging across Amazonia sensu lato (that includes the Amazon Basin and the Guiana Shield). Based on long-term (8–30 year) inventory data from 13 experimentally-disturbed sites (*Sist et al., 2015*) across Amazonia (*Figure 1— figure supplement 1*), 133 permanent forest plots (175 ha in total) that cover a large gradient of disturbance intensities (ACS losses ranging from 1% to 71%) were used to model the trajectory of those

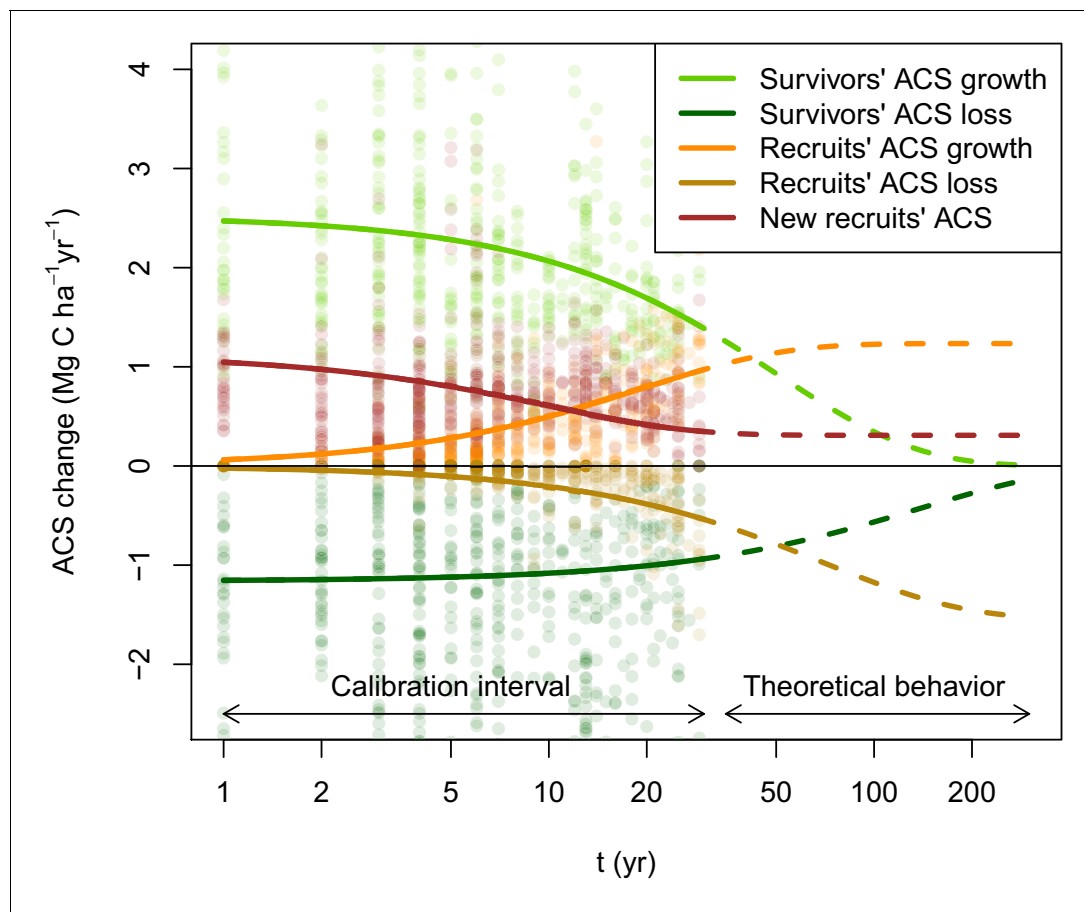

**Figure 1.** Post-disturbance annual ACS changes of survivors and recruits in 133 Amazonian selectively logged plots. Data is available between the year of minimum ACS ($t = 0$) and $t = 30$ years. ACS changes are: recruits' ACS growth (orange), recruits' ACS loss (gold), new recruits' ACS (red), survivors' ACS growth (light green) and survivors' ACS loss (dark green). Thick solid lines are the maximum-likelihood predictions (for an average plot, when all covariates are null), and dashed lines are the model theoretical behaviour. New recruits' ACS, recruits' ACS growth, and recruits' ACS loss converge over time to constant values. A dynamic equilibrium is then reached: ACS gain from recruitment and recruits' growth compensate ACS loss from recruits' mortality. Survivors' ACS growth and loss. decline over time and tend to zero when all initial survivors have died.

The following figure supplement is available for figure 1:

**Figure supplement 1.** Experimental sites location, each site being composed of permanent forest plots varying in logging intensities, census length (colour) and total area (size).

post-disturbance ACS changes (*Figure 1*) in a comprehensive Bayesian framework. We quantify the effect of pre-disturbance ecosystem characteristics [the site's average pre-logging ACS (*acs*0) and the relative difference between each plot and *acs*0 as a proxy of forest maturity (*dacs*)], disturbance intensity [percentage of pre-logging ACS lost (*loss*)], and interactions with the environment [annual precipitation (*prec*), seasonality of precipitation (*seas*), and soil bulk density (*bd*)] (*Figure 2*) on the rates at which post-disturbance ACS changes converge to a theoretical steady state (as in *Figure 1*, see Materials and methods for more details). With global maps of ACS (*Avitabile et al., 2016*), climatic conditions (*Hijmans et al., 2005*) and soil bulk density (*Nachtergaele et al., 2008*), we upscale our results to Amazonia (sensu lato) and elaborate predictive maps of potential ACS changes over 10 years under the hypothesis of a 40% ACS loss, which is a common disturbance intensity after conventional logging in Amazonia (*Blanc et al., 2009*; *Martin et al., 2015*; *West et al., 2014*). Summing these ACS changes over time gives the net post-disturbance rate of ACS accumulation. Disentangling ACS recovery into demographic processes and cohorts is essential to reveal mechanisms underlying ACS responses to disturbance and to make more robust predictions of ACS recovery compared to an all-in-one approach (see Appendix).

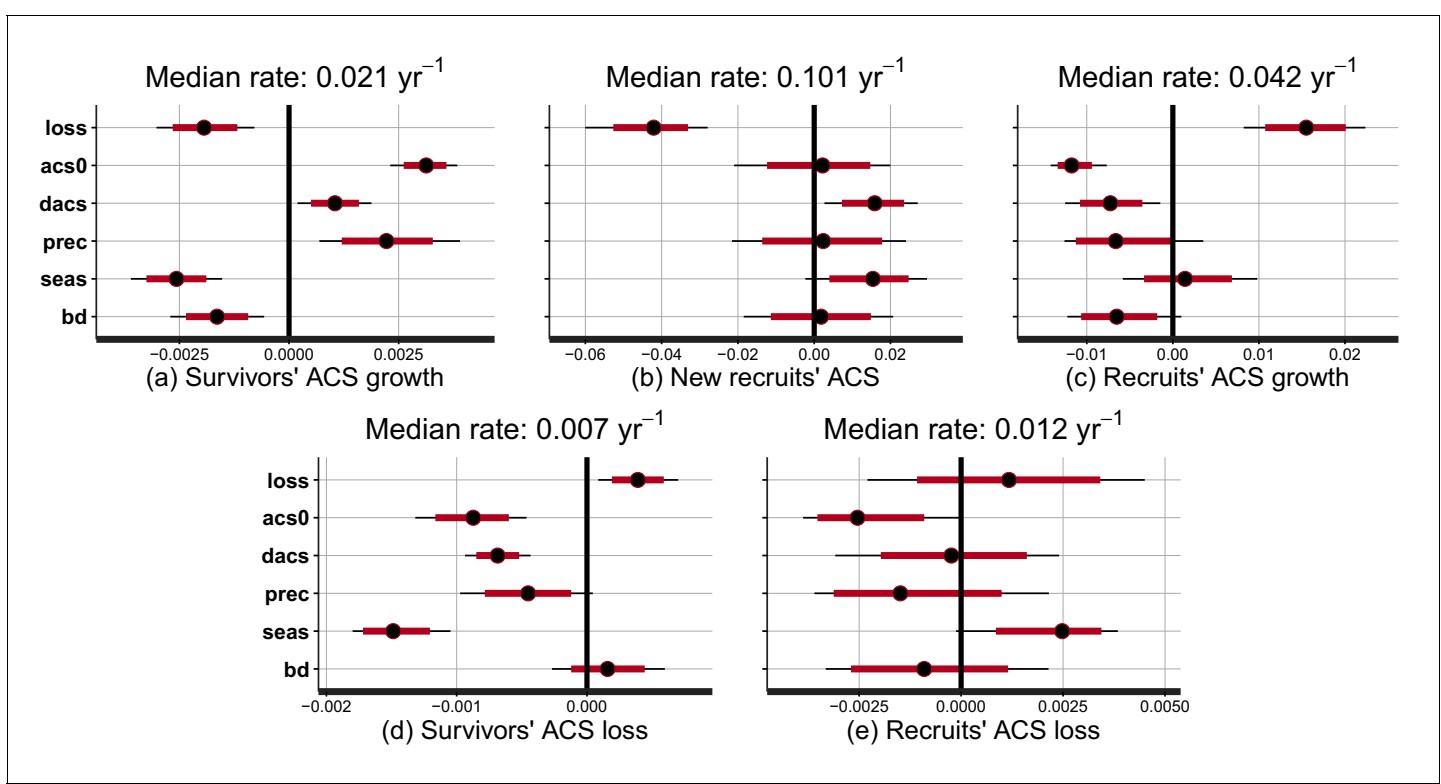

**Figure 2.** Effect of covariates on the rate at which post-disturbance ACS changes converge to a theoretical steady state (in $yr^{-1}$). Covariates are : disturbance intensity (*loss*) , i.e. the proportion of initial ACS loss; mean site's ACS (*acs*0), and relative forest maturity, i.e. pre-logging plot ACS as a % of *acs*0 (*dacs*); annual precipitation (*prec*); seasonality of precipitation (*seas*), soil bulk density (*bd*). Covariates are centred and standardized. Red and black levels are 80% and 95% credible intervals, respectively. The median rate is the prediction of the convergence rate for an average plot (when all covariates are set to zero). Negative covariate values indicate slowing and positive values indicate accelerating rates. (a) Survivors' ACS growth. (b) New recruits' ACS. (c) Recruits' ACS growth. (d) Survivors' ACS loss. (e) Recruits' ACS loss.

The following source data and figure supplement are available for figure 2:

**Source data 1.** Parameters posterior distribution.

**Figure supplement 1.** Fitted vs observed values of cumulative ACS changes (Mg C ha$^{-1}$).

## Results

### Local variations of ACS changes

At a given site, variations of post-logging ACS changes are explained with the disturbance intensity (*loss*) and the relative forest maturity (*dacs*). At high disturbance intensity (positive *loss*) as well as in relatively immature forests (negative *dacs*), ACS gain from recruits is high: recruitment decreases slowly (*Figure 2b* and *Figure 3b*) and recruits' growth increases rapidly (*Figure 2c* and *Figure 3c*). In the same conditions of high disturbance intensity, survivors' ACS growth is lower in the first years following logging than for low disturbance intensities, but declines slowly (*Figure 2a* and *Figure 3a*). Disturbance intensity and relative forest maturity have a weak effect on ACS loss from both survivors and recruits (*Figures 2d,e* and *3d,e*). Overall, net ACS change stays high longer at high disturbance intensity (*Figure 3f*).

### Regional variations of ACS changes

Variations of post-logging ACS changes between sites are explained with the mean ACS of each site (*acs0*), climatic conditions [annual precipitation (*prec*), seasonality of precipitation (*seas*)] and the soil bulk density (*bd*). Contribution of survivors' growth to ACS recovery declined slowly in sites with low *acs0* and high water stress (low precipitation, high seasonality and high bulk density) (*Figure 2a*). Survivors' ACS loss showed the opposite pattern (*Figure 2d*) except in apparent response to high seasonality of precipitation (*seas*) that slowed the post-disturbance rates of decline of both ACS growth and loss. Despite slower recruits' ACS growth in sites with high pre-logging ACS (*acs0*), no other regional covariate had significant effects on recruits' ACS changes (*Figure 2b,c and e*).

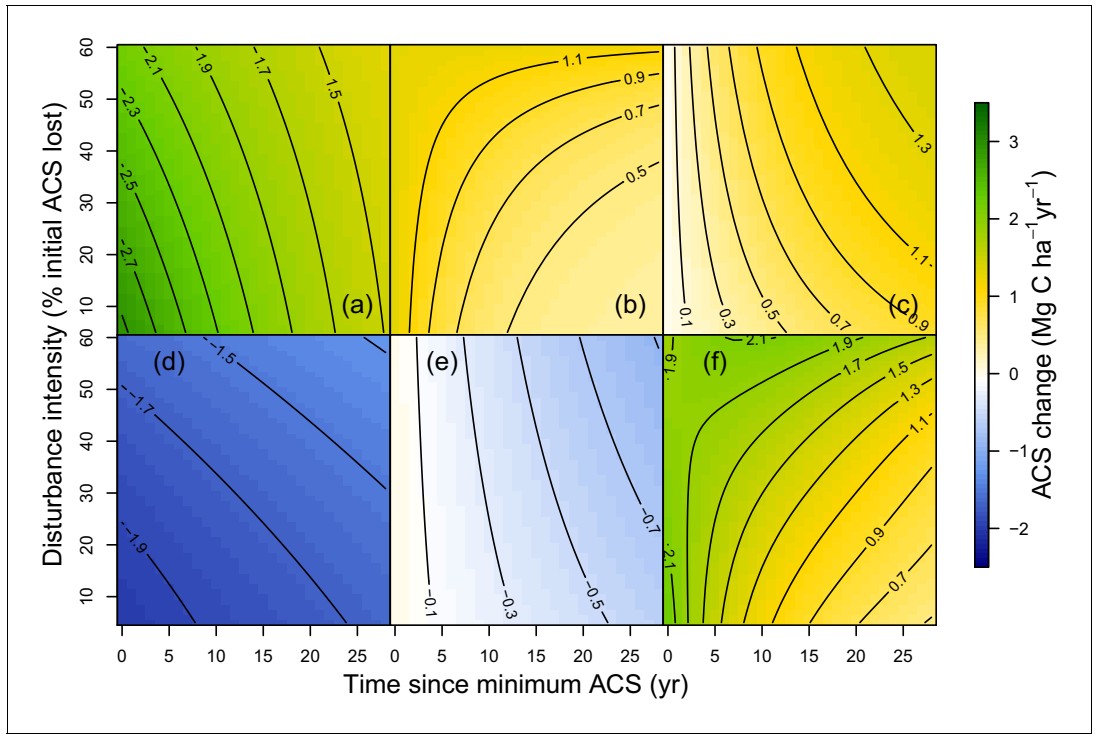

**Figure 3.** Predicted effect of disturbance intensity on ACS changes along time in an Amazonian-average plot. (a) Survivors' ACS growth. (b) New recruits' ACS. (c) Recruits' ACS growth. (d) Survivors' ACS loss. (e) Recruits' ACS loss. (f) Net ACS change. The net ACS change is the sum of all five ACS changes. ACS changes were calculated with all parameters set to their maximum-likelihood value and covariates (except standardized disturbance intensity *loss*) set to 0. Time since minimum ACS varies from 0 to 30 year (i.e. the calibration interval) and disturbance intensity ranges between 5% and 60% of initial ACS loss.

## Prediction maps

While no significant environmental effects were detected for recruits' ACS changes (*Figures 2* and *4*), the survivors showed a highly structured regional gradient: (i) ACS gain from survivors' ACS growth is high in the west and in the Guiana Shield, but low in the south (*Figure 4a*), whereas (ii) survivors' ACS loss is low in the south and in the Guiana Shield but high in the west (*Figure 4d*). To illustrate how these regional differences will be critical for future ACS across Amazonia, we developed a map of net ACS recovery over the first 10 years after a 40% ACS loss by integrating the sum of ACS change predictions through time (*Figure 5*). Across the region, net ACS recovery over the first ten years after a 40% ACS loss is predicted to be $17 \pm 7$ Mg C ha$^{-1}$, with higher values in the west and in the Guiana Shield (*Figure 5a*). The uncertainty in predictions was low to medium (coefficient of variation under 40%) in 82% of the mapped area, and high (coefficient of variation above 50%) in 5% of the mapped area (*Figure 5b*).

Four areas (*Figure 5a*) were selected to represent four contrasted cases of net ACS recovery in time (*Figure 6*): two areas, northwestern Amazonia and the Guiana Shield, with high ACS accumulation ($21 \pm 3$ Mg C ha$^{-1}$ over 10 year), one intermediate area, central Amazonia ($15 \pm 1$ Mg C ha$^{-1}$

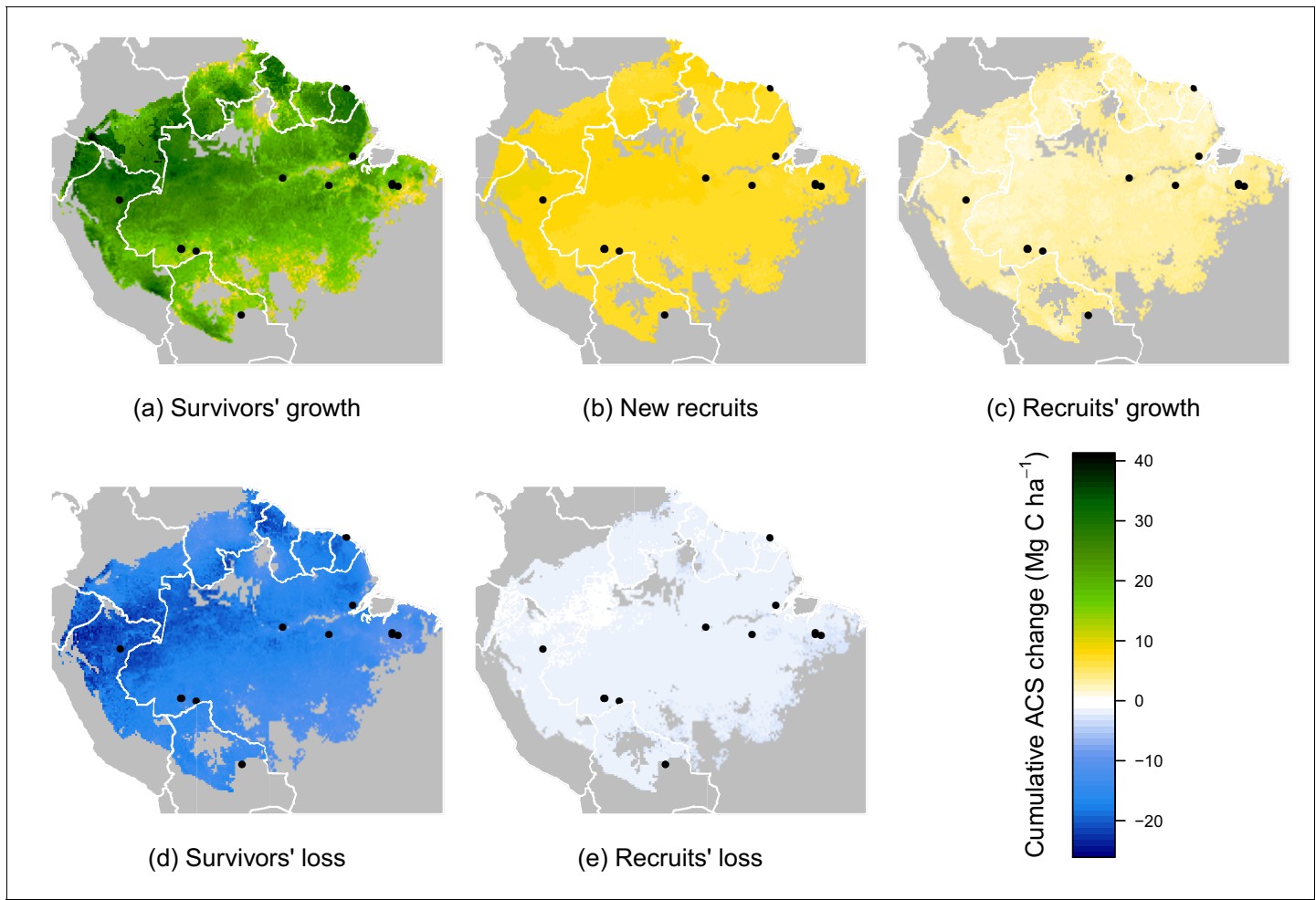

**Figure 4.** Predicted cumulative ACS changes (Mg C ha$^{-1}$) over the first 10 year after losing 40% of ACS. Extrapolation was based on global rasters: topsoil bulk density from the Harmonized global soil database (*Nachtergaele et al., 2008*), Worldclim precipitation data (*Hijmans et al., 2005*) and biomass stocks from Avitabile et al. map (*Avitabile et al., 2016*). Cumulative ACS changes are obtained by integrating annual ACS changes through time. We here show the median of each pixel. Top graphs are ACS gain and bottom graphs are ACS loss. (a) ACS gain from survivors' growth. (b) ACS gain from new recruits. (c) ACS gain from recruits' growth. (d) ACS loss from survivors' mortality. (e) ACS loss from recruits' mortality. Black dots are the location of our experimental sites. Survivors' ACS changes (a and d) show strong regional variations unlike to recruits' ACS changes (b,c and e).

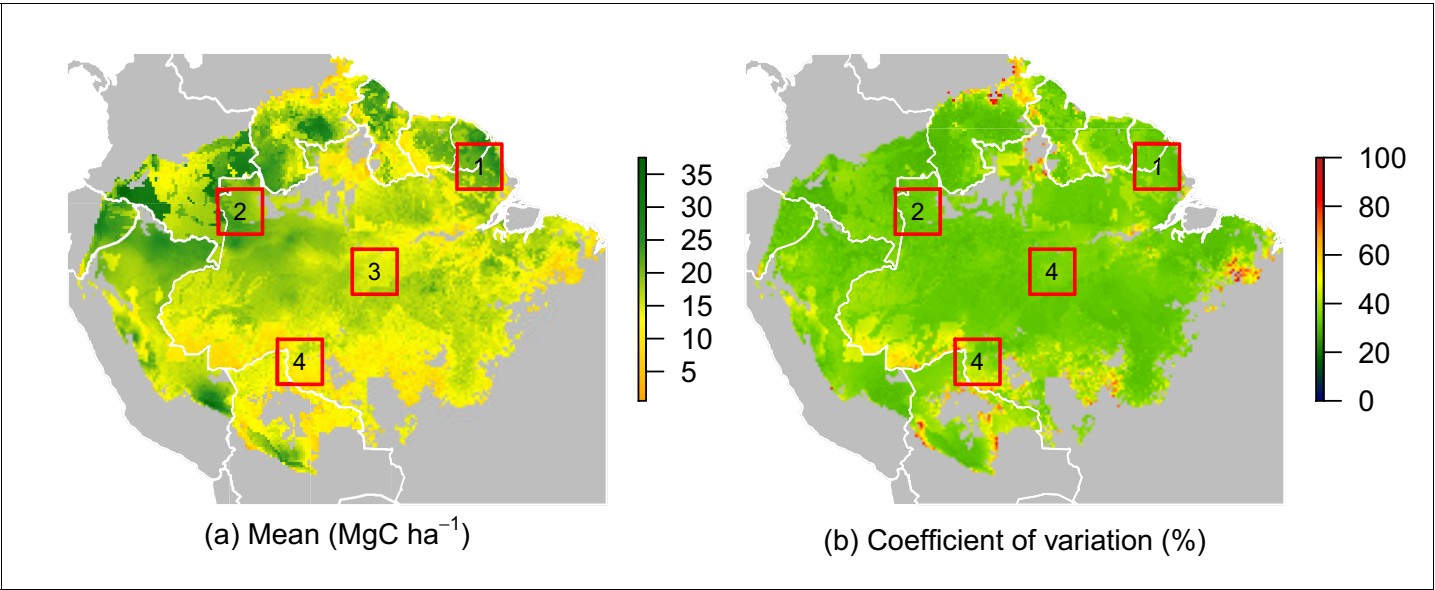

**Figure 5.** Predicted net ACS recovery over the first 10 year after losing 40% of pre-logging ACS. (a) median predictions. (b) coefficient of variation (per pixel). Four areas were arbitrarily chosen to illustrate four different geographical behaviours: (1) the Guiana Shield and (2) northwestern Amazonia are two areas with high ACS recovery; the Guiana Shield has higher initial ACS and slower ACS dynamics whereas northwestern Amazonia has lower initial ACS and faster ACS dynamics. (3) central Amazonia has intermediate ACS recovery. (4) southern Amazonia has low ACS recovery.

over 10 year) and one area with low ACS accumulation, southern Amazonia ($12 \pm 3$ Mg C ha$^{-1}$ over 10 year). Survivors' contribution to the sum of ACS gains (recruitment and growth) over the first 10 years after disturbance was $71 \pm 4\%$ in the Guiana Shield, $71 \pm 2\%$ in the west; $63 \pm 4\%$ in central Amazonia and $55 \pm 6\%$ in the south. Predicted net ACS recovery (*Figure 5*) and survivors' ACS growth (*Figure 4a*) are highly correlated: $\rho = 0.90$ (Pearson's correlation coefficient).

## Discussion

Contrasting post-disturbance ACS dynamics were detected among the western Amazon, Guiana Shield, and southern Amazon (*Figure 4*). (i) In the western Amazon, environmental stress is reduced due to fertile soils and abundant, mostly non-seasonal precipitation, but forests are prone to frequent and sometimes large-scale wind-induced disturbances (*Espírito-Santo et al., 2014*). Such conditions of low stress and high disturbance tend to favor fast-growing species with rapid life cycles (*He et al., 2013*), which results in fast ACS gain and loss from survivors even after the logging disturbance (*Figures 4a,d* and *6*). (ii) Forests of the Guiana Shield are generally dense and grow on nutrient-poor soils (*Quesada et al., 2012*), where wood productivity is highly constrained by competition for key nutrients, especially phosphorus and nitrogen (*Santiago, 2015*; *Mercado et al., 2011*). The short duration pulse of nutrients released from readily decomposed stems, twigs and leaves of trees damaged and killed by logging may thus explain the substantial but limited-duration increase in growth of survivors on these nutrient-poor soils (*Figure 6*). Yet post-disturbance ACS loss from survivors' mortality decreases slowly in the Guiana Shield (*Figure 6*). This is consistent with the low mortality rates and the high tree longevity reported in old-growth forests of this region (*Phillips et al., 2004*). (iii) In the southern Amazon, high seasonal water stress is the main constraint on ACS recovery (*Wagner et al., 2016*). Stress-tolerant trees are generally poor competitors (*He et al., 2013*) and this may explain the slow ACS changes of survivors in this region (*Figures 4a,d* and *6*). Finally, Central Amazonia is a transition zone for the main environmental and biotic gradients found in Amazonia: (1) a competition gradient between dense and nutrient-poor northeastern forests and nutrient-rich western forests; (2) an environmental gradient between northern wet forests and southern drier forests (*Quesada et al., 2012*).

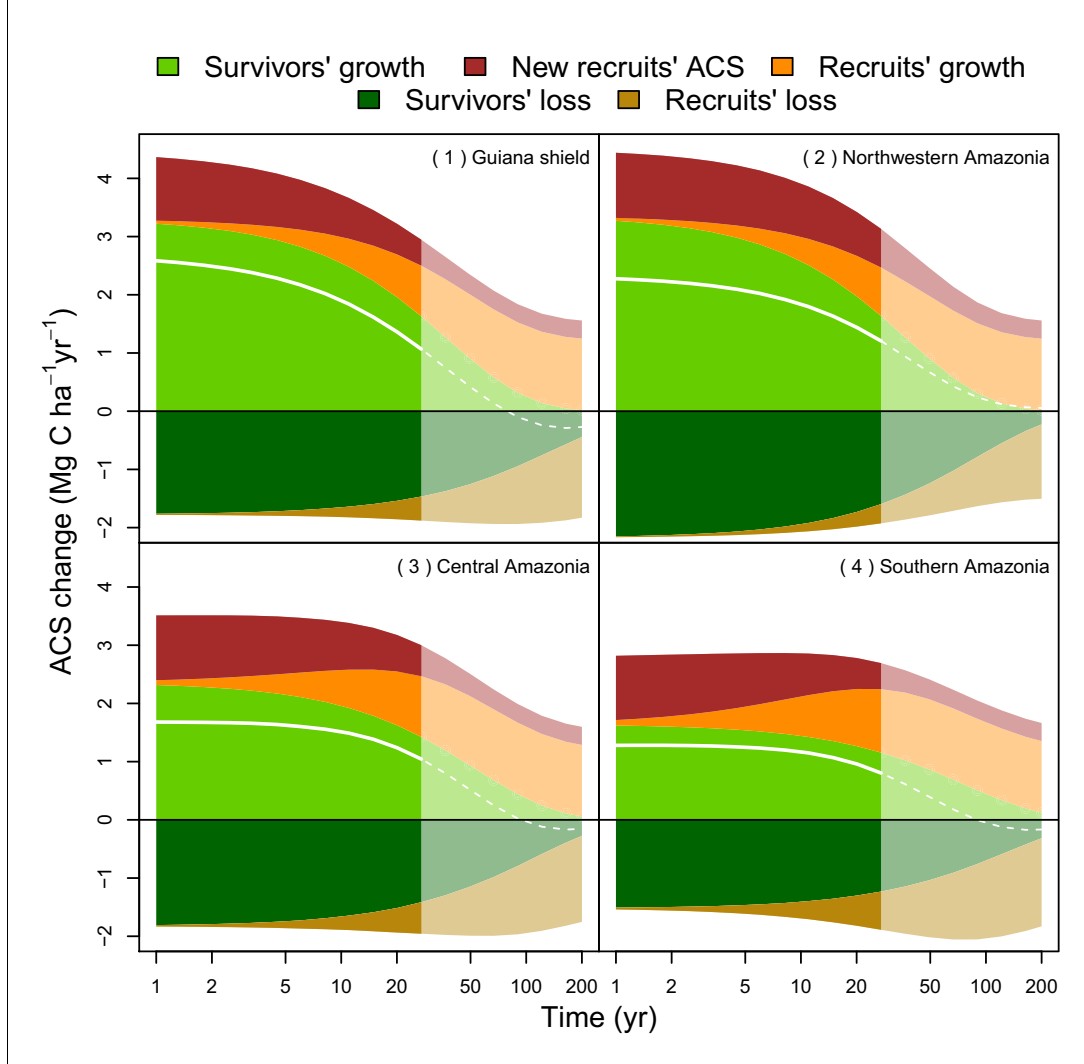

**Figure 6.** Predicted contribution of annual ACS changes in ACS recovery in four regions of Amazonia (*Figure 5*). The white line is the net annual ACS recovery, i.e. the sum of all annual ACS changes. Survivors' (green) and recruits' (orange) contribution are positive for ACS gains (survivors' ACS growth, new recruits' ACS and recruits' ACS growth) and negative for survivors' and recruits' ACS loss. Areas with higher levels of transparency and dotted lines are out of the calibration period (0–30 year). In the Guiana Shield and in nothwestern Amazonia, high levels of net ACS recovery are explained by large ACS gain from survivors' growth. Extrapolation was based on global rasters: topsoil bulk density from the Harmonized global soil database (*Nachtergaele et al., 2008*), precipitation data from Worldclim (*Hijmans et al., 2005*) and biomass stocks from Avitabile et al. (*Avitabile et al., 2016*) map.

Across Amazonia, survivors contribute most to post-disturbance ACS recovery. In regions where survivors' ACS gain is high (west and northeast), net ACS recovery is also high: annual ACS recovery is between 1 and 3 Mg C ha$^{-1}$ yr$^{-1}$ in the first 10 year after logging (*Figure 6*), lower than in Amazonian secondary forests (3–5 Mg C ha$^{-1}$ yr$^{-1}$ in the first 20 year after abandonment of land use [*Poorter et al., 2016*]). Recruits, for their part, have very low geographical variations in post-logging ACS changes: 10 years after the disturbance they are predicted to store similar amounts of ACS almost everywhere in Amazonia. Nevertheless, small trees with DBH <20 cm have not been accounted for in our study and may play an important role in post-logging ACS changes. The 10–20 cm DBH size class contains as much as 14% of total ACS and may be highly dynamic in some Amazonian forests (*Vieira et al., 2004*). Because of the slow tree growth rates in Amazonia (*Vieira et al., 2005*; *Herault et al., 2010*), many trees will not reach the 20 cm DBH threshold 10 years after logging: the effects of the 10–20 cm DBH stratum on post-logging ACS changes are likely to be missed

in sites with less than 10 years of measurements (e.g. Peteco, Ecosilva, Iracema, Cumaru) and should be studied, together with the natural regeneration, in the future.

At the stand level, high disturbance intensities reduce survivors' ACS: survivors' ACS growth is consequently lower (*Figure 3a*), resulting in lower net ACS change during the first 10 years of the recovery period (*Figure 3f*). High disturbance intensities as well as relatively low forest maturity alleviate competition, and this is probably why ACS contributions from recruits remain high for longer (*Figure 2b*) in such enhanced growth conditions (*Herault et al., 2010*). In the first years after logging, net ACS recovery depends little on disturbance intensity (*Figure 3f*), but recovery is predicted to last longer in heavily logged forests. In immature forests, intense self-thinning (*Swaine et al., 1987*) may explain fast ACS losses from survivors' mortality (*Figure 2d*).

In the tropics, reduced-impact logging techniques (RIL; [*Putz et al., 2008*]) are promoted to reduce collateral damage to residual stands and biodiversity. Our results reveal that lower disturbance intensities, as a direct consequence of the employment of RIL techniques, could increase survivors' ACS growth and slow down their ACS loss. Given that government specified minimum cutting cycles are short, e.g. 35 year in the Brazilian Amazon (*Blaser et al., 2011*), and that many commercial species are slow-growing and dense-wooded (*Dauber et al., 2005*; *Wright et al., 2010*), available timber stocks for the next cutting cycle will be comprised mostly of survivors. Attention should be taken to high harvest intensities and/or substantial incidental damage due to poor harvesting practices that diminish stocks of survivors, even if they promote recruitment. Most trees that recruit are fast-growing pioneers that are favored by disturbance but are vulnerable to water stress (*Bonal et al., 2016*) and competition (*Valladares and Niinemets, 2008*), and because their height is lower than in mature forests (*Rutishauser et al., 2016*), they might have reduced carbon sequestration potential. With ongoing climate change and increased frequencies and intensities of droughts in Amazonia (*Malhi et al., 2008*), betting on recruits to store C in forests disturbed by selective logging might thus be a risky gamble.

In this study, we focus on one type of disturbance: selective logging. Because of its economic value and implications for forest management, selective logging is a long-studied human disturbance in tropical forests, and the data gathered by the TmFO network are unique in terms of experiment duration and spatial extent. We nevertheless believe that our study gives clues on the regional differences in Amazonian forests response to large ACS losses induced by other disturbances (e.g. droughts, fire) that are expected to increase in frequency with ongoing global changes (*Bonal et al., 2016*).

## Materials and methods

### Site description

Our study includes data from thirteen long-term (8–30 year) experimental forest sites located in the Amazon Basin and the Guiana Shield (*Figure 1—figure supplement 1*). Sites meet the following criteria: (i) located in tropical forests with mean annual precipitation above 1000 mm; (ii) a total censused area above 1 ha; (iii) at least one pre-logging census and (iv) at least two post-logging censuses. For each site, we extracted annual precipitation and seasonality of precipitation data from WorldClim (RRID:SCR_010244) (*Hijmans et al., 2005*), topsoil bulk density data from the Harmonized World Soil database (*Nachtergaele et al., 2008*), and the synthetic climatic index from Chave et al. (*Chave et al., 2014*), using in all cases the highest resolution data available (30 arc-seconds). For one of our sites (La Chonta, see *Figure 1—figure supplement 1*), field measurements of precipitation (mean = 1580 mm yr$^{-1}$) differed substantially from WorldClim data (1032 mm yr$^{-1}$): in this particular case we used the measured value and adjusted the synthetic climatic index (E) in the allometric equation (*Chave et al., 2014*) accordingly. Sites' data is available at Dryad Digital Repository (*Piponiot et al., 2016*).

### ACS computation

In all plots, diameter at breast height (DBH) of trees >20 cm DBH were measured, and trees were identified to the lowest taxonomic level: to the species level (75%) when possible, or to the genus level (15%); 10% of trees were not identified. To get the wood density, we applied the following standardized protocol to all sites: (i) trees identified to the species level were assigned the

corresponding wood specific gravity value from the Global Wood Density Database (GWDD, doi:10.5061/dryad.234/1) (*Zanne et al., 2009*); (ii) trees identified to the genus level were assigned a genus-average wood density; (iii) trees with no botanical identification or that were not in the GWDD were assigned the site-average wood density. The aboveground biomass (AGB) was estimated with the allometric equations from Chave et al. (*Chave et al., 2014*). Biomass was assumed to be 50% carbon (*Penman et al., 2003*). The ACS of every tree $i$ was then computed as follows:

$$
\hat{ACS}_i = exp\Big( -1.803 - 0.976 \times E + 0.976 \times ln(WD_i) +
$$
$$
+ 2.673 \times ln(DBH_i) - 0.0299 \times ln(DBH_i^2)\Big) \times 0.5
$$

(1)

where $WD_i$ and $DBH_i$ are the specific wood density and diameter at breast height of the tree $i$ and $E$ is the synthetic climatic index (*Chave et al., 2014*). The ACS changes data that was generated is available at Dryad Digital Repository (*Piponiot et al., 2016*).

## The recovery period

After logging, plot ACS decreases rapidly until it reaches its minimum value (*acsmin*) a few years later. This transition point determines the beginning $t_{min} = t_0$ of the recovery period. *acsmin* was estimated as the minimum ACS in the 4 years following logging activities. Because our focus is on post-logging ACS recovery, we did not include in our analysis plots where the minimum ACS value was not reached within the 4 years after logging, either because the logging activity did not affect the plot or because there were other sources of disturbance long after logging (fire, road opening, silvicultural treatments).

## ACS changes computation

For each plot $j$ and census $k$, with $t_k$ the time since the beginning of the recovery period $t_0$, we define 5 ACS changes : new recruits' ACS ($Rr_{j,k}$) is the ACS of all trees <20 cm DBH at $t_{k-1}$ and ≥20 cm DBH at $t_k$; recruits' ACS growth ($Rg_{j,k}$) is the ACS increment of living recruits between $t_{k-1}$ and $t_k$ ; recruits' ACS loss ($Rl_{j,k}$) is the C in recruits that die between $t_{k-1}$ and $t_k$; survivors' ACS growth ($Sg_{j,k}$) is the ACS increment of living survivors between $t_{k-1}$ and $t_k$; survivors' ACS loss ($Sl_{j,k}$) is the ACS of survivors that die between $t_{k-1}$ and $t_k$. ACS gains ($Sg$, $Rr$, $Rg$) are positive and ACS losses ($Sl$, $Rl$) are negative. Instantaneous ACS changes are subject to stochastic variation over time: because we are less interested in year-to-year variations than in long-term ACS trajectories, we modelled cumulative ACS changes instead of annual ACS changes. Cumulative ACS changes (Mg C ha$^{-1}$) were defined as follows:

$$
cChange_{j,k} = \sum_{m=0}^{k} Change_{j,m}
$$

(2)

where $j$ is the plot, $t_k$ the time since $t_0$ (yr) and *Change* is the annual ACS change (Mg C ha$^{-1}$ yr$^{-1}$), either recruits' ACS ($Rr$), recruits' ACS growth ($Rg$), recruits' ACS loss ($Rl$), survivors' ACS growth ($Sg$), or survivors' ACS loss ($Sl$).

## Covariates

To model ACS changes, we chose six covariates : (1) *loss* disturbance intensity, i.e. percentage of initial ACS loss; (2) *acs0* mean ACS of the site; (3) *dacs* relative ACS of the plot, as a % of *acs0*; (4) *prec* annual precipitation; (5) *seas* precipitation seasonality; (6) *bd* topsoil bulk density. To give equivalent weight to all covariates, we centred and standardized them in order to have a mean of zero and a standard deviation of one over all observations. The uncertainty associated with ACS covariates (*loss*, *acs0*, *dacs*) is less than 10% (*Chave et al., 2014*). Climatic covariates (annual precipitation *prec* and precipitation seasonality *seas*) were extracted from Worldclim rasters (RRID:SCR_010244). Error in Worldclim precipitation data was estimated to be <10 mm in Amazonia (*Hijmans et al., 2005*). There is no information on the uncertainty on topsoil bulk density but we expect it to be higher than the uncertainty on other covariates, due to measurement (*De Vos et al., 2005*) and interpolation methods (*Hendriks et al., 2016*).

## Survivors' model

Survivors' cumulative ACS changes are null at $t = 0$ (by definition). When all survivors are dead, their ACS changes stop: annual ACS changes become null and cumulative ACS changes reach a constant/finite limit. We decided to model survivors' cumulative ACS growth $cSg$ and ACS loss $cSl$ as:

$$cS_{i,j,k} \sim \mathcal{N}\left(\alpha_j^S \times \left(1 - exp(-\beta_j^S \times t_k)\right), (\sigma_E^S)^2\right) \tag{3}$$

where $j$ is the plot, $t_k$ is the time since $t_0$, $S$ is either $Sg$ or $Sl$. $\alpha_j^S$ is the finite limit of the cumulative ACS change and $\beta_j^S$ the rate at which the cumulative ACS change converges to this limit. By choosing an exponential kernel, we assume that survivors' ACS change at $t_k$ is proportional to survivors' ACS change at $t_k - 1$.

Because $\alpha_j^S$ values are expected to vary among plots, they are modelled with the following distribution:

$$\alpha_j^S \sim \mathcal{N}\left(\alpha_0^S, (\sigma_\alpha^S)^2\right) \tag{4}$$

Parameter $\beta_j^S$ is the rate at which survivors' ACS change (from growth or mortality) on plot $j$ converges to a finite limit after the disturbance: it reflects the response rapidity of survivors' ACS changes to disturbance. Because we are interested in predicting variations in $\beta_j^S$ ($S$ is either $Sg$ or $Sl$), we expressed $\beta_j^S$ as a function of covariates:

$$\beta_j^S = \beta_0^S + \sum_{l=1}^{6} (\lambda_l^S \times V_{j,l}) \tag{5}$$

where $\sum_{l=1}^{6}(\lambda_l^S \times V_{j,l})$, is the effect of covariates ($V_{j,l}$) on the post-logging rate $\beta_j$. Covariates are centred and standardized and are (1) *loss* : disturbance intensity, i.e. percentage of initial ACS loss; (2) *acs0* : mean ACS of the site; (3) *dacs* relative ACS of the plot, as a % of *acs0*; (4) *prec* annual precipitation; (5) *seas* precipitation seasonality; (6) *bd* topsoil bulk density.

When all survivors in plot $j$ are dead, all the C gained by their growth ($cSg_{j,\infty} = \alpha_j^{Sg}$) plus their initial ACS ($acsmin_j$) will have been lost ($cSl_{j,\infty} = \alpha_j^{Sl}$). We thus added the following constraint to each plot $j$:

$$\alpha_j^{Sl} + \alpha_j^{Sg} + acsmin_j = 0 \tag{6}$$

with $\alpha_j^{Sg}, \alpha_j^{Sl}$ the finite limits of survivors' cumulative ACS growth and ACS loss respectively, and $acsmin_j$ the ACS of the plot $j$ at $t_{min} = t_0$.

## Recruits' model

When survivors are all dead, recruits will constitute the new forest. We made the assumption that the ACS of this new forest will reach a dynamic equilibrium: recruits' annual ACS changes are expected to converge to constant values (that are however prone to small inter-annual variations), with ACS gains compensating ACS losses. Because there are no recruits yet at $t_0$, recruits' annual ACS growth ($Rg$) and ACS loss ($Rl$) are zero, and progressively increase to reach their asymptotic values. Recruits' annual ACS growth and ACS loss can be thus modelled with the function:

$$f(t; \alpha, \beta) = \alpha \times \left(1 - exp(-\beta \times t)\right) \tag{7}$$

where $t$ is the time since the beginning of the recovery period. In the same logic as survivors' cumulative ACS change, $\alpha$ is the asymptotic value of recruits' annual ACS change (Mg C ha$^{-1}$ yr$^{-1}$), and $\beta$ is the rate at which this asymptotic value is reached.

Contrary to recruits' annual ACS growth and ACS loss, the ACS of new recruits ($Rr$) is high at $t_0$ because of the competition drop induced by logging, but then progressively decreases to reach its asymptotic value. We modelled it with the following function:

$$f(t; \alpha, \beta, \eta) = \alpha \times \left(1 + \eta \times exp(-\beta \times t)\right) \tag{8}$$

where $t$ is the time since logging. The parameter $\eta$ was added to allow annual recruited ACS to be higher than $\alpha$ at $t_0$.

As stated before, we chose to model cumulative ACS changes instead of annual ACS changes. The general model for recruits' cumulative ACS changes is deduced by integrating annual ACS changes from $t_0$ to $t_k$:

$$cR_{i,j,k} \sim \mathcal{N}\left(\alpha_i^R \times \left(t_k + \eta \times \frac{1 - exp(-\beta_j^R \times t_k)}{\beta_j^R}\right), (\sigma_E^R)^2\right) \tag{9}$$

where $i$ is the site, $j$ is the plot, $t_k$ is the time since $t_0$ is either $Rr, Rg$ or $Rl$. When $R$ is $Rg$ or $Rl$, $\eta = -1$; when $R$ is $Rr$, $\eta > 0$.

Once the forest reaches a new dynamic equilibrium, recruits' annual ACS changes should depend mostly on each site's characteristics: we expect there to be more inter-site than intra-site variation in recruits' asymptotic ACS changes $\alpha^R$. This is why we use one value $\alpha_i^R$ per site $i$, and model it as follows:

$$\alpha_i^R \sim \mathcal{N}\left(\alpha_0^R; (\sigma_\alpha^R)^2\right) \tag{10}$$

When the dynamic equilibrium is reached, annual ACS gain (growth and recruitment) compensates annual ACS loss (mortality). We thus added the following constraint for every site $i$:

$$\alpha_i^{Rr} + \alpha_i^{Rg} + \alpha_i^{Rl} = 0 \tag{11}$$

With the same logic as for survivors, we are interested in predicting variation in $\beta^R$. Given that we use one value $\alpha_i^R$ per site $i$ (i.e. all plots in one site $i$ have the same value for $\alpha_i^R$), we chose to take into account the inter-plot variability as follows:

$$\beta_j^R \sim \mathcal{N}\left(\beta_0^R + \sum_{l=1}^{6}(\lambda_l^R \times V_{j,l}), (\sigma_\beta^R)^2\right) \tag{12}$$

## Inference

Bayesian hierarchical models were inferred through MCMC methods using an adaptive form of the Hamiltonian Monte Carlo sampling (*Carpenter et al., 2015*). Each observation was given a weight proportional to the size of the plot. Codes were developed using the R language (RRID:SCR_ 001905) (*R Developement Core Team, 2015*) and the Rstan package (*Carpenter et al., 2015*). A detailed list of priors is provided in *Table 1*.

## Prediction maps

Maps were obtained with the following steps: (i) spatially-explicit covariates are extracted at the resolution of 30 arc-second from: the pan-tropical carbon map of Avitabile et al. for pre-disturbance aboveground carbon stocks (*Avitabile et al., 2016*); WorldClim (RRID:SCR_010244) (*Hijmans et al., 2005*) for annual precipitation and seasonality of precipitation, and the Harmonized World Soil database (*Nachtergaele et al., 2008*) for topsoil bulk density; (ii) disturbance intensity is set to 40% of pre-logging ACS loss, which is a common value for disturbance intensity after conventional logging in Amazonia (*West et al., 2014*; *Blanc et al., 2009*; *Martin et al., 2015*) , and the relative forest maturity *dacs* is set to zero; (iii) parameters are drawn from their previously calibrated distribution; (iv) to simulate random effects, all five parameters ($\alpha$) are taken from their distribution $\mathcal{N}(\alpha_0, \sigma_\alpha^2)$; (v) for every pixel, we estimate the five cumulative ACS changes ($cSg$, $cSl$, $cRr, cRg, cRl$) 10 years after the 40% ACS loss, given the parameters value and the pixel covariates values extracted from global rasters. Steps (iii) to (v) are repeated 200 times and summary statistics are calculated for every pixel. Because a significant part of our sites have experiment duration lower than 10 years (*Figure 1—figure supplement 1*), we are less confident in Amazonian-wide predictions after that 10 year period. Maps were elaborated under the R statistical software (RRID:SCR_001905) (*R Developement Core Team, 2015*).

**Table 1.** List of priors used to infer ACS changes in a Bayesian framework. Models are : ($Sg$) survivors' ACS growth, ($Sl$) survivors' ACS loss, ($Rr$) new recruits' ACS, ($Rg$) recruits' ACS growth, ($Rl$) recruits' ACS loss. $\lambda_{loss}$ is the parameter relative to the covariate *loss* (logging intensity).

| Model | Parameter | Prior | Justification |
|---|---|---|---|
| $Sg$ | $\alpha_j^{Sg}$ | $\mathcal{U}[25, 250]$ | On average 100 survivors/ha storing 0.25 to 2.5 MgC each |
| $Sg$ | $\beta_j^{Sg}$ | $\mathcal{U}[0.015, 0.04]$ | $75 < t_{0.95}^{Sg}{}^* < 200$ yr |
| $Sl$ | $\beta_j^{Sl}$ | $\mathcal{U}[0.006, \beta^{Sg}]$ | $t_{0.95}^{Sg} < t_{0.95}^{Sl}{}^* < 500$ yr |
| $Rr$ | $\alpha_i^{Rr}$ | $\mathcal{U}[0.1, 1]$ | Range of observed values in TmFO control plots |
| $Rr$ | $\beta_j^{Rr}$ | $\mathcal{U}[0.006, 0.6]$ | $5 < t_{0.95}^{Rr}{}^* < 500$ yr |
| $Rr$ | $\eta$ | $\mathcal{U}[0, 3]$ | $Rr(t = 0) < 3 \times Rr(t = \infty)$ |
| $Rg$ | $\alpha_i^{Rg}$ | $\mathcal{U}[0.5, 3]$ | Range of observed values in Amazonia (*Johnson et al., 2016*) |
| $Rg$ | $\beta_j^{Rg}$ | $\mathcal{U}[0.006, 0.15]$ | $20 < t_{0.95}^{Rg}{}^* < 500$ yr |
| $Rl$ | $\beta_j^{Rl}$ | $\mathcal{U}[0.003, 0.06]$ | $50 < t_{0.95}^{Rl}{}^* < 1000$ yr |
| All models M[†] | $\lambda_{loss}^M$ | $\mathcal{U}[-\beta^M, \beta^M]$ | Avoid multicollinearity problems |
| All models M[†] | $(\lambda_l^M)_{l \neq loss}$ | $\mathcal{U}[-\frac{\beta^M}{4}, \frac{\beta^M}{4}]$ | Avoid multicollinearity problems |

[*] $t_{0.95} = \frac{ln(20)}{\beta}$ is the time when the ACS change has reached 95% of its asymptotic value.

[†] M is one of the five models: either $Sg$, $Sl$, $Rr$, $Rg$, $Rl$.

## Acknowledgements

We are in debt with all technicians and colleagues who helped setting up the plots and collecting data over years. Without their precious work, this study would have not been possible and they may be warmly thanked here. We are grateful to CIRAD, the GFclim project (FEDER 2014–2020, Project GY0006894) and the Sao Paulo Research Foundation (FAPESP: 2013/16262–4 and 2013/50718–5) for financial support. This study was partially funded by an Investissement d'Avenir grant of the ANR (CEBA: ANR-10-LABEX-0025) and carried out in the framework of the Tropical managed Forests Observatory (TmFO), supported by the Sentinel Landscape program of CGIAR (Consultative Group on International Agricultural Research) - Forest Tree and Agroforestry Research Program.

## Additional information

### Funding

| Funder | Grant reference number | Author |
|---|---|---|
| Agence Nationale de la Recherche | ANR-10-LABEX-0025 | Camille Piponiot Bruno Hérault |
| Fundação de Amparo à Pesquisa do Estado de São Paulo | FAPESP: 2013/16262-4 and 2013/50718-5 | Edson Vidal |
| European Regional Development Fund | FEDER 2014-2020, GY0006894 | Camille Piponiot Bruno Hérault |

The funders had no role in study design, data collection and interpretation, or the decision to submit the work for publication.

### Author contributions

CP, ER, Conception and design, Analysis and interpretation of data, Drafting or revising the article; PS, BH, Conception and design, Acquisition of data, Analysis and interpretation of data, Drafting or revising the article; LM, CPdA, MF, MVNd'O, CRdS, Acquisition of data, Analysis and interpretation of data, Drafting or revising the article; MP-C, Conception and design, Acquisition of data, Drafting

or revising the article; FEP, AS, CB, Analysis and interpretation of data, Drafting or revising the article; NA, MG, ENHC, ARR, KEdS, EDS, EV, TAPW, Acquisition of data, Drafting or revising the article

### Author ORCIDs

Camille Piponiot, http://orcid.org/0000-0002-3473-1982

Eurídice N Honorio Coronado, http://orcid.org/0000-0003-2314-590X

Bruno Hérault, http://orcid.org/0000-0002-6950-7286

## Additional files

### Major datasets

The following dataset was generated:

| Author(s) | Year | Dataset title | Dataset URL | Database, license, and accessibility information |
|---|---|---|---|---|
| Piponiot C, Sist P, Mazzei L, Peña-Claros M, Putz F, Rutishauser E, Shenkin A, Ascarrunz N, de Azevedo C, Baraloto C, França M, Guedes M, Honorio Coronado E, d'Oliveira MVN, Ruschel AR, da Silva KE, Doff Sotta E, de Souza CR, Vidal E, West TAP, Hérault B | 2016 | Data from: Post-disturbance carbon recovery in Amazonian forests | http://dx.doi.org/10.5061/dryad.rc279 | Available at Dryad Digital Repository under a CC0 Public Domain Dedication |

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

## Appendix

### The importance of a process-based approach

We here study C recovery dynamics with a demographic process-based approach, i.e. by segregating ACS changes into cohorts (survivors and recruits) and demographic processes (growth, recruitment, mortality), as opposed to an all-in-one model in which only the ecosystem net ACS change is modelled, without examination of demographic processes.

To compare the goodness of fit of the two approaches (all-in-one and process-based), we calibrated an all-in-one model with our data and compared the accuracy of its predictions with the process-based predictions reported in this study. The all-in-one model was written as follows:

$$C_{j,k} \sim \mathcal{N}\left((acs0_j - acsmin_j) \times (1 - exp(-\beta_j^C \times t_k)), (\sigma_E^C)^2\right) \tag{13}$$

where $C_{j,k}$ (MgC ha$^{-1}$) is the total C accumulation $t_k$ years after the disturbance in plot $j$ is the ACS lost by logging and $\beta_j^C$ is the rate at which the plot ACS returns to its pre-logging ACS. We took into account the effect of covariates and dependencies for $\beta_j^C$ :

$$\beta_j^C \sim N\left(\beta_0^C + \sum_{l=1}^{6}(\lambda_l^C \times V_{j,l}); (\sigma_\beta^C)^2\right) \tag{14}$$

The process-based model made better predictions (RMSE = 0.24) than the all-in-one model (RMSE = 0.31). In some sites, for example Paracou (black diamonds in **Appendix—figure 1**), there is a clear bias in the all-in-one model predictions: C accumulation is overestimated at the beginning of the recovery period and underestimated towards the end. This bias may be due to the non-adequacy of the negative exponential curve in the classic all-in-one model ( **Appendix—figure 2a**) to the C recovery observed in experimental plots (**Rutishauser et al., 2015**). The process-based model does not predict a constant instantaneous C accumulation rate (**Appendix—figure 2b**), and is thus more accurate.

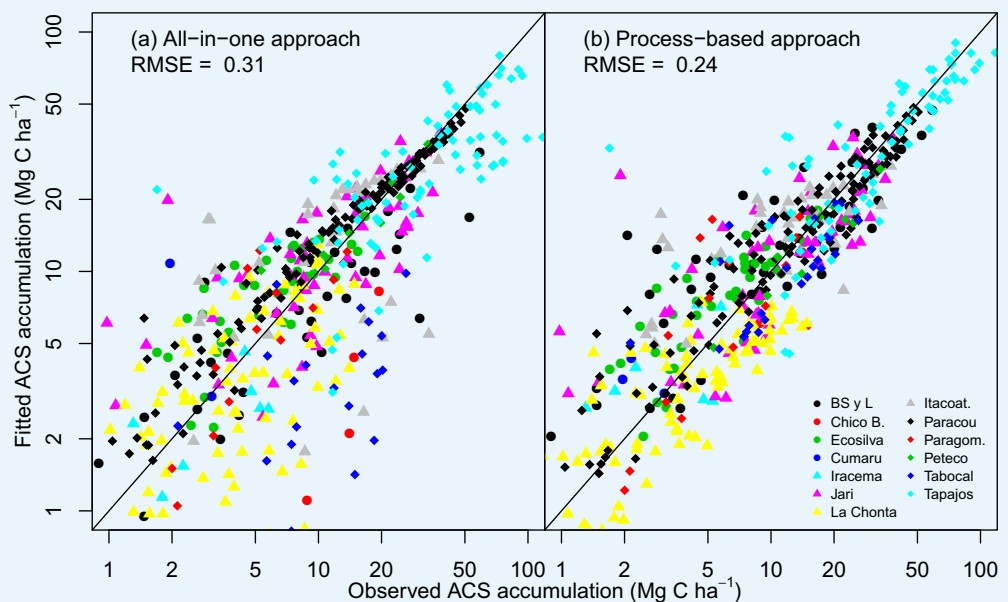

**Appendix 1—figure 1.** Observed vs fitted values of net ACS accumulation (MgC ha$^{-1}$). (**a**) Fitted values from the all-in-one model. (**b**) Fitted values from the process-based model (right). Net ACS accumulation is the sum of cumulative ACS changes (gain and loss). Each combination of a colour and shape is specific to a site. The closer the dots are to the x=y line, the better the prediction.

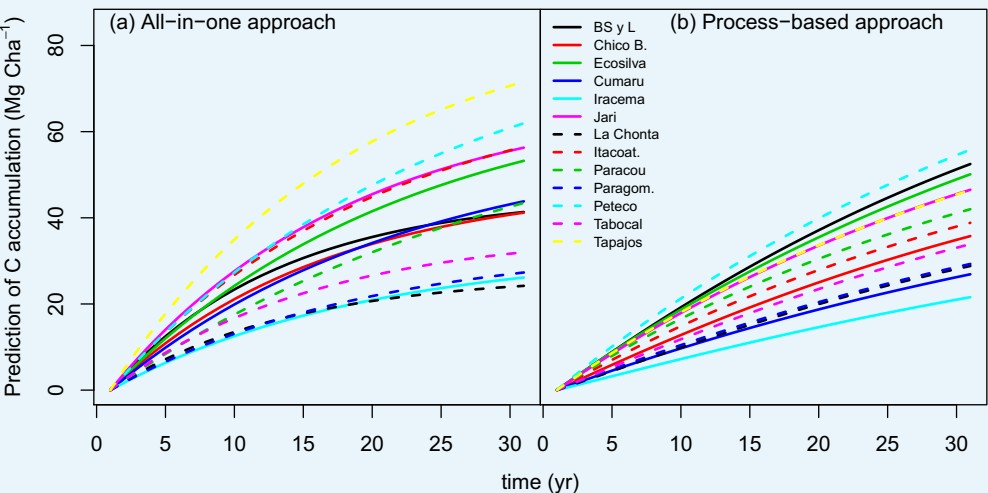

**Appendix 1—figure 2.** Predicted trajectories of net ACS accumulation (MgC ha$^{-1}$) per site with (**a**) the all-in-one model and (**b**) the process-based model.

