## [Decision Letter]

Thank you for submitting your article "Post-disturbance carbon recovery in Amazonian forests" for consideration by *eLife*. Your article has been reviewed by two peer reviewers, and the evaluation has been overseen by a Reviewing Editor and Ian Baldwin as the Senior Editor. The reviewers have opted to remain anonymous.

The reviewers have discussed the reviews with one another and the Reviewing Editor has drafted this decision to help you prepare a revised submission; this decision relies heavily on the two reviewers' comments, as they largely agreed with each other.

Title: The authors use a very broad term "disturbance" when they actually deal with only a single type of disturbance: selective logging. They need to change the title to reflect their study more accurately. (Suggestion: C recovery following selective logging.)

Summary:

Piponiot et al. synthesize data from a number of sites across the Amazon Basin to assess carbon (C) recovery dynamics following selective logging. The novelty of their study is in the observation-driven approach that distinguishes growth and mortality in recruitments and survivors as controls on the rate of biomass recovery after logging. They linked regional differences in climate, soil and initial biomass with differences in the relative importance growth and mortality to build a model that can provide Amazon-wide predictions of post-logging accumulation of above ground C stocks. This study has potential implications for forest management as well as indicating the importance of selective logging on the C balance of the Amazon region.

Essential revisions:

The reviewers identified 5 main points that need to be addressed. These do not require major re-analysis of the data, but do require that the authors are clearer about what they have done and what the consequences of some of their decisions are for their overall results.

1) The Results and Discussion refer to disturbance intensity (for example in the last paragraph of the Discussion) but Results and Discussion were restricted to a single 'scenario' (ten years after losing 40% of pre-disturbance ACS). Could you add a figure and discussion on, for example, how fitted growth and mortality rates vary as a function of disturbance intensities? Alternatively, the post-disturbance dynamics could be plotted for a single region but for contrasting disturbance intensities. Logging intensity is too important and interesting to be not discussed in more detail.

2) Discussion, last paragraph: 'available timber stocks […]' is a statement with potentially far reaching consequences for forest management. Can you support/quantify this statement by making use of your model? Both reviewers would be interested in the authors could comment on, or even better, include more results on this topic. Reviewer 1 suggested a number of interesting questions: how many cutting cycles can one make before the forest is degraded? If the forest is degraded how long would it take for the recruits to recover the biomass to 50% of the pre-harvest level? What would be the time in between two cutting cycles to justify calling the forest management 'sustainable'? The whole point of selective logging should be to extract wood while at the same time enhance biomass recovery by maintaining high stand-level productivity. Any results analysis that could provide a more quantitative perspective on this issue (including regional differences across the Amazon) would be exciting.

While not insisting on additional analysis by the authors, some comment on how their study can contribute to these questions would be useful.

3) The authors show that variations in ACS changes of recruits are smaller than that of survivors. In fact, this is one of the main results of their study. Although large trees contain an important fraction of the stand aboveground biomass in tropical forests (Slik et al. 2013), small trees contribute significantly to the species diversity pool and biomass stocks/balance. This is especially important in Central and Western Amazon forests, where large trees are less abundant and account for a lower proportional biomass/basal area than in Eastern Amazon forests (Vieira et al. 2004). Therefore, a highly dynamic stratum (trees < 20 cm DBH) is missing in the analyses. This stratum will define the structure and floristic composition of the future forests, directly related to timber volume and quality for the next harvesting cycles. Moreover, given the low growth rates of Amazon species (Silva et al. 2002; Hérault et al. 2010) and the minimum DBH considered in this study (only trees ⩾ 20 cm DBH were monitored), trees that were considered 'recruits' are in fact more likely to be 'survivors' (i.e. already there before logging). This can be especially true in the first 5-10 yrs following logging (e.g. Iracema and Peteco), where trees established after logging probably did not reach the 20 cm DBH threshold. Apart from the distinction in size (only given in the last section of the paper), the authors do not provide clear evidence that 'recruits' survived the disturbance.

As the main results and conclusions of the paper rely on the importance/contribution of the cohorts, these limitations and their implications need to be addressed.

4) The authors need to be clearer about how they compute ACS and make sure some of their methods are stated early in the text. Specifically, reviewer 2 notes that the limitation to trees of > 20 cm DBH (and is this greater than or equal to, or greater than; see subsection “ACS changes computation”) has consequences for the results of the paper and should be mentioned as part of the study.

Also, the assignment of plot-level wood density to calculate ACS is problematic, especially when the plot is recovering from disturbance. While pioneer species can have wood density values lower than 0.35 g cm^-3^ (e.g. *Tapirira obtusa* and *Jacaranda copaia*), late-successional species can have wood density values three times higher (> 1 g cm^-3^; e.g. *Swartzia corrugata* and *Licania laxiflora*). When only trees with DBH ⩾ 20 cm are monitored, and a given plot has a higher abundance of 'survivors' (i.e. large and relatively old trees), the biomass of pioneer trees can be overestimated in an unknown order of magnitude. In contrast, in a plot dominated by pioneer species (e.g. those where logging was more intense), this effect will be the opposite. These two situations affect analyses on the importance of cohorts (i.e. survivors and recruits) and demographic processes (i.e. *Rr, Rg, RI, Sg* and *SI*).

At a minimum, the authors some assessment of how assumptions of wood density (as opposed to using local allometric equations that rely solely on DBH) could affect their conclusions.

5) Floristic data were acquired at the sites and mentioned in the Methods, but no results or summary was presented. It is not clear what the authors mean with 'trees were identified to the lowest taxonomic level' (i.e. 'species level')? Was there potentially important information that was not mentioned? Please add some information about the number of species, genera and botanical families recorded in each site, or give a citation where such information can be found. It is particularly of interest if floristic diversity varies among the 13 sites.

---

## [Author Response]

*The reviewers have discussed the reviews with one another and the Reviewing Editor has drafted this decision to help you prepare a revised submission; this decision relies heavily on the two reviewers' comments, as they largely agreed with each other.*

*Title: The authors use a very broad term "disturbance" when they actually deal with only a single type of disturbance: selective logging. They need to change the title to reflect their study more accurately. (Suggestion: C recovery following selective logging.)*

The title has been changed to “Carbon recovery dynamics following disturbance by selective logging in Amazonian forests”.

We agree that our data only covers one type of disturbance: selective logging. It seems however important to us to stress out the fact that because selective logging is a long-studied disturbance (because of its implications for forest management and its economic value), we have a unique dataset (in terms of experiment duration and spatial extent) to study the effect of disturbance on Amazonian tropical forests. We thus believe that our study could give clues on the regional differences in Amazonian forests response to other disturbance types (i.e. drought-induced large mortalities, fire). This is why we added a section in the Discussion (last paragraph) to emphasize this point.

*[…] Essential revisions:*

*The reviewers identified 5 main points that need to be addressed. These do not require major re-analysis of the data, but do require that the authors are clearer about what they have done and what the consequences of some of their decisions are for their overall results.*

*1) The Results and Discussion refer to disturbance intensity (for example in the last paragraph of the Discussion) but Results and Discussion were restricted to a single 'scenario' (ten years after losing 40% of pre-disturbance ACS). Could you add a figure and discussion on, for example, how fitted growth and mortality rates vary as a function of disturbance intensities? Alternatively, the post-disturbance dynamics could be plotted for a single region but for contrasting disturbance intensities. Logging intensity is too important and interesting to be not discussed in more detail.*

Very important point. Thank you. The Figure 3 was added: it clearly illustrates the predicted effect of disturbance intensity (with all other covariates set to 0) on all 5 ACS changes, and on the net ACS change (i.e. the sum of all 5 ACS changes). The new figure is introduced in the Results section (subsection “Local variations of ACS changes”) and discussed later (Discussion, third paragraph). Thank you for this nice suggestion. It helps clarify our results and improve the manuscript.

*2) Discussion, last paragraph: 'available timber stocks [...]' is a statement with potentially far reaching consequences for forest management. Can you support/quantify this statement by making use of your model? Both reviewers would be interested in the authors could comment on, or even better, include more results on this topic. Reviewer 1 suggested a number of interesting questions: how many cutting cycles can one make before the forest is degraded? If the forest is degraded how long would it take for the recruits to recover the biomass to 50% of the pre-harvest level? What would be the time in between two cutting cycles to justify calling the forest management 'sustainable'? The whole point of selective logging should be to extract wood while at the same time enhance biomass recovery by maintaining high stand-level productivity. Any results analysis that could provide a more quantitative perspective on this issue (including regional differences across the Amazon) would be exciting.*

*While not insisting on additional analysis by the authors, some comment on how their study can contribute to these questions would be useful.*

We agree that the issues related to timber stocks raised by the reviewers are of great interest for forest managers and policy makers. Unfortunately, many of those questions are beyond the reach of our study:

We deal with carbon stocks of all trees > 20 cm DBH that cannot be directly converted into timber stocks (standard definition: volume of commercial trees with DBH > 50 cm): we thus cannot use our model to predict timber stock recovery. To call forest management ‘sustainable’, we should at least be sure that timber stocks recover at the end of a cutting cycle and this cannot be inferred from our “carbon-stock” model.

Because all the TmFO experimental plots have been logged only once (and this is equally true for many forest plots across the tropics), we have no data to predict the effect of multiple cutting cycles.

We can however provide partial answers to some of the questions raised:

When all covariates (except disturbance intensity *loss*) are set to 0, i.e. for an average Amazonian plot, recruits reach 50% of pre-logging ACS after 96, 88, 78 and 59 years when the initial ACS lost (disturbance intensity) is 5%, 10%, 20% and 40% respectively. We are not very confident in these figures because they are way out of our calibration interval (0-30 years). That’s why we prefer not to discuss those predicted numbers in the manuscript.

We also looked at predictions of the recovery time, i.e. the time taken to recover part of the C initially lost. This is a question that we have often asked ourselves because of its implications in defining ‘sustainable’ cutting cycles. Because our model has been calibrated with absolute C recovery rates (or ACS changes) and not relative recovery rates (% of ACS lost), we don’t believe that it is well-fitted to predict relative recovery. As an example, we plotted the 95% credible interval of time taken to recover 50% of ACS lost in the 4 regions highlighted in the manuscript (Guiana Shield, Northwestern Amazonia, Southern Amazonia, Central Amazonia), with varying disturbance intensities (Figure 9). As shown previously (Rutishauser et al., 2015), recovery time clearly increases with disturbance intensity: that’s nothing new. Regional variations in recovery time are mostly driven by differences in initial ACS (e.g. northwestern Amazonia and the Guiana shield have similar absolute recovery rates but differ in initial ACS and thus in recovery times). At each disturbance intensity, 95% credible intervals of all 4 regions overlap and it is thus uneasy to have clear conclusions.

Author response image 1.Time to recover 50% of initial ACS per region and disturbance intensity.**DOI:**
http://dx.doi.org/10.7554/eLife.21394.015

*3) The authors show that variations in ACS changes of recruits are smaller than that of survivors. In fact, this is one of the main results of their study. Although large trees contain an important fraction of the stand aboveground biomass in tropical forests (Slik et al. 2013), small trees contribute significantly to the species diversity pool and biomass stocks/balance. This is especially important in Central and Western Amazon forests, where large trees are less abundant and account for a lower proportional biomass/basal area than in Eastern Amazon forests (Vieira et al. 2004). Therefore, a highly dynamic stratum (trees < 20 cm DBH) is missing in the analyses. This stratum will define the structure and floristic composition of the future forests, directly related to timber volume and quality for the next harvesting cycles. Moreover, given the low growth rates of Amazon species (Silva et al. 2002; Hérault et al. 2010) and the minimum DBH considered in this study (only trees ⩾ 20 cm DBH were monitored), trees that were considered 'recruits' are in fact more likely to be 'survivors' (i.e. already there before logging). This can be especially true in the first 5-10 yrs following logging (e.g. Iracema and Peteco), where trees established after logging probably did not reach the 20 cm DBH threshold. Apart from the distinction in size (only given in the last section of the paper), the authors do not provide clear evidence that 'recruits' survived the disturbance.*

*As the main results and conclusions of the paper rely on the importance/contribution of the cohorts, these limitations and their implications need to be addressed.*

We agree that a 20 cm DBH threshold is unusually high, but imposed by the forest plot data, and that this should be made clearer and discussed. We added the DBH threshold in the Introduction (third paragraph) to make it more explicit. This issue is discussed in the second paragraph of the Discussion.

*4) The authors need to be clearer about how they compute ACS and make sure some of their methods are stated early in the text. Specifically, reviewer 2 notes that the limitation to trees of > 20 cm DBH (and is this greater than or equal to, or greater than; see subsection “ACS changes computation”) has consequences for the results of the paper and should be mentioned as part of the study.*

We added a mention to the DBH threshold in the Introduction and discuss the implications of not accounting for trees < 20 cm DBH (Discussion, second paragraph).

*Also, the assignment of plot-level wood density to calculate ACS is problematic, especially when the plot is recovering from disturbance. While pioneer species can have wood density values lower than 0.35 g cm^-3^ (e.g. Tapirira obtusa and Jacaranda copaia), late-successional species can have wood density values three times higher (> 1 g cm^-3^; e.g. Swartzia corrugata and Licania laxiflora). When only trees with DBH ⩾ 20 cm are monitored, and a given plot has a higher abundance of 'survivors' (i.e. large and relatively old trees), the biomass of pioneer trees can be overestimated in an unknown order of magnitude. In contrast, in a plot dominated by pioneer species (e.g. those where logging was more intense), this effect will be the opposite. These two situations affect analyses on the importance of cohorts (i.e. survivors and recruits) and demographic processes (i.e. Rr, Rg, RI, Sg and SI).*

*At a minimum, the authors some assessment of how assumptions of wood density (as opposed to using local allometric equations that rely solely on DBH) could affect their conclusions.*

Over our data, 75% of individual trees (>20 cm DBH) were allocated a wood density to the species level, 15% were assigned a genus-average wood density and only 10% were undetermined and were given the site average wood density (we added determination levels for every site on the Dryad table “Sites description”). Species-level wood density variation is mostly explained at the genus level (Jérôme Chave, Muller-Landau, Baker, Easdale, & ter Steege, 2006): only trees for which we have neither the species nor the genus wood density could be problematic, but they represent a minority of trees. We thus think that our assumptions on wood density don’t influence significantly ACS estimations. To explore that issue, we work with the only site where indetermination levels are higher than 5% (the Paracou site).

We tested the effect of botanical indetermination on cumulative ACS changes in Paracou (Figure 2) as follows:

1) We set wood density of all undetermined trees to either the site’s average wood density (as is done in the study, 0.692 in Paracou), either to 0.4 (lower bound) or to 0.9 (higher bound), and plotted the 5 cumulative ACS changes (i.e. the data that is used in the inference) for all 9 logged plots in Paracou.

2) Only survivors’ loss (dark green) is significantly affected by the low botanical identification (Figure 10).

3) Given that the uncertainty range doesn’t increase with time after the initial 10 years period, this means that this uncertainty comes from trees that were not identified during the site set up and died in the first decade, so that they could not be formally identified later.

4) It is unlikely that the mean density of those unidentified trees (survivors present in the forest before logging) differs significantly from the site’s mean wood density.

This is why we believe that even in Paracou, the level of determination is not critical for our estimation of ACS changes.

Author response image 2.Effect of botanical indetermination on cumulative ACS changes in the 9 Paracou forest plots.Wood density of all undetermined trees is set to 0.4 (lower bound), 0.9 (higher bound), or the plot average wood density (dashed lines): the latter is the method used in the study. Cumulative ACS changes are then calculated. Cumulative ACS changes (MgC/ha) are: survivors’ ACS growth (light green), survivors’ ACS loss (dark green), new recruits’ ACS (red), recruits’ growth (orange), recruits’ ACS loss (ocher).**DOI:**
http://dx.doi.org/10.7554/eLife.21394.016

*5) Floristic data were acquired at the sites and mentioned in the Methods, but no results or summary was presented. It is not clear what the authors mean with 'trees were identified to the lowest taxonomic level' (i.e. 'species level')? Was there potentially important information that was not mentioned? Please add some information about the number of species, genera and botanical families recorded in each site, or give a citation where such information can be found. It is particularly of interest if floristic diversity varies among the 13 sites.*

We tried to make clearer how trees were identified, and especially the level of determination (subsection “ACS computation”). We add some information about the species richness (ha^-1^) range recorded at each site Dryad table “Sites description” (Dryad Digital Repository: http://dx.doi.org/10.5061/dryad.rc279).